# Chain of Thought in Order: Discovering Learning-Friendly Orders for Arithmetic

## Abstract

The chain of thought, i.e., step-by-step reasoning, is one of the fundamental mechanisms of Transformers. While the design of intermediate reasoning steps has been extensively studied and shown to critically influence performance, the ordering of these steps has received little attention, despite its significant effect on the difficulty of reasoning. This study addresses a novel task of unraveling the chain of thought—reordering decoder input tokens into a learning-friendly sequence for Transformers, for learning arithmetic tasks. The proposed pipeline first trains a Transformer on a mixture of target sequences arranged in different orders and then identifies benign orders as those with fast loss drops in the early stage. As the search space grows factorially in sequence length, we propose a two-stage hierarchical approach for inter- and intra-block reordering. Experiments on four order-sensitive arithmetic tasks show that our method identifies a learning-friendly order out of a few billion candidates. Notably, on the multiplication task, it recovered the reverse-digit order reported in prior studies.

## 1 Introduction

Autoregressive generation is central to the success of the Transformer (Vaswani et al., 2017) in reasoning tasks, which leads to many successes of the end-to-end learning of arithmetic and hard symbolic computations, such as (Lample & Charton, 2020; Charton, 2022; Kera et al., 2024; 2025; Alfarano et al., 2024; Wenger et al., 2022; Li et al., 2023a;b). The autoregressive nature makes each reasoning step conditioned on the preceding context, and careful design of intermediate reasoning steps, such as *chain of thought* (Wei et al., 2022), guides the model's reasoning toward the final answer of the target problem. For example, it has been known that learning the parity function—the prediction of the parity of the input bit string—is challenging (Shalev-Shwartz et al., 2017; Hahn & Rofin, 2024). However, Kim & Suzuki (2025) recently has shown that the step-by-step prediction of the parity of the first $k$ bits with $k = 1, 2, \ldots$, makes the learning successful.

One important yet underexplored aspect is the order of the chain of thought—not only which steps to include, but also the order in which they are arranged can greatly impact learning. For example, Shen et al. (2023) has shown that Transformers learn multiplication of two integers with better generalization to larger integers (i.e., those with more digits) when the product is predicted from least to most significant digits (cf. Figure 1). While this particular case can be explained by the carries, which flow from least to most significant digits, a systematic way of determining a learning-friendly order of the chain of thought remains unknown.

In this study, we address a new task of reordering decoder input tokens into a learning-friendly order for better learning of arithmetic tasks. Exploiting the observation that neural networks tend to learn from easy to hard instances during training (Arpit et al., 2017; Forouzesh & Thiran, 2024; Swayamdipta et al., 2020), we train a Transformer on a mixture of target sequences in different orders and identify those that lead to a faster loss drop in the early stages of training. To better handle longer sequences, we propose a two-stage hierarchical approach, where the global stage finds block-level orders, while the local stage reorders tokens within each block.

The experiments demonstrate that the proposed method successfully reorders the target sequences. We designed three arithmetic tasks that are relatively easy to compute with the (input and) target

sequence in the forward order, but not with other orders. Starting from random orders, the proposed method succeeds up to thirteen tokens (i.e., $13! > 6 \times 10^9$ permutations), increasing the success rate of arithmetic computation from approximately $10\%$ to $100\%$. We also applied our method to the multiplication task in (Shen et al., 2023) and successfully rediscovered the reverse orders.

Our contributions are summarized as follows:

- We address a novel task, unraveling the chain of thought. This aims at discovering a learning-friendly order of decoder input tokens, thereby making the learning more successful for in-distribution samples and generalizable to out-of-distribution samples.

- We propose a method that efficiently determines learning-friendly orders from the loss profile at the early stage of training. Empirically, this filters a few thousand candidates in a single epoch, and combined with a hierarchical strategy, the best order can be found out of a few billion candidates.

- We introduce order-sensitive arithmetic tasks using non-injective maps, with which one can evaluate reordering methods. Our extensive experiments present that the proposed method successfully discovers learning-friendly orders and rediscover the previously reported the learning-friendly order in the multiplication task.

## 2 RELATED WORK

**Transformers for mathematical tasks.** Transformers have recently been applied to mathematical problem-solving with encouraging results. (Lample & Charton, 2020) has demonstrated that a Transformer can carry out integral calculus with a high success rate, opening the possibility that sequence-to-sequence models can handle algebraic tasks. Since that study, applications have expanded to arithmetic (Charton, 2022), linear algebra (Charton, 2022), computational algebra (Kera et al., 2024; 2025), and coding theory, as well as cryptography (Wenger et al., 2022; Li et al., 2023a;b). One reason behind these successes is the autoregressive generation scheme. Although theory has suggested that learning high-sensitive functions such as parity is difficult (Hahn & Rofin, 2024), recent work achieved a high success rate on parity tasks by applying a chain of thought prompting (Wei et al., 2022; Kojima et al., 2022; Chen et al., 2023; Yao et al., 2023; Zhang et al., 2024) to arithmetic and by exploiting the generated output tokens effectively (Kim & Suzuki, 2025). Positional encoding is also crucial for arithmetic problems; prior work (Jelassi et al., 2023) has shown that relative-position and abacus-style embeddings improve generalization to out-of-distribution data. These studies collectively show that task-specific representations and positional encodings strongly influence performance. In particular, prior work (Shen et al., 2023) analyzed in detail how digit order affects multiplication success rate and demonstrated that generating digits from the least significant position upward raises the success rate; however, the ordering was chosen heuristically rather than by an automated procedure. Systematic optimization of the output order itself in arithmetic tasks remains unaddressed. This study is the first to exploratively optimize the output-sequence permutation for each task, automatically discovering a learning-friendly target order.

**Easy-to-hard learning dynamics in deep neural networks.** The observation that deep neural networks can be trained even on randomly assigned labels—while still achieving excellent generalization on real data—led to a line of research into how models adapt to data during training (Zhang et al., 2017). (Arpit et al., 2017) has experimentally shown that networks first pick up simple regularities between inputs and labels and only later transition to memorizing harder, noise-like examples. More broadly, deep neural networks are known to learn easy instances in a dataset before gradually fitting the more difficult ones; in image domains, this behavior is often referred to as spectral bias (Rahaman et al., 2019). This property is now widely exploited in curriculum learning (Jiang et al., 2018; Han et al., 2018; Baldock et al., 2021) and data-quality control (Swayamdipta et al., 2020). For example, integrating each sample's learning curve can reveal mislabeled data (Forouzesh & Thiran, 2024). Most prior work, however, analyzes such dynamics by injecting noise directly into the target labels themselves. In contrast, this study focuses on the ordering of the target sequences. The dataset is rearranged with multiple permutation matrices, and the model is trained on these reordered versions to investigate how sequence order affects learning.

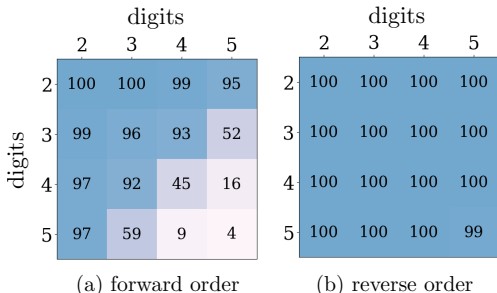

(a) forward order  (b) reverse order

Figure 1: Success rates for the multiplication of two integers. Matrix rows and columns indicate the number of digits in each operand. Evaluation is conducted with 100 samples for each digit position. (a) The model is trained to output from the most significant digit. (b) The model is trained to output from the least significant digit.

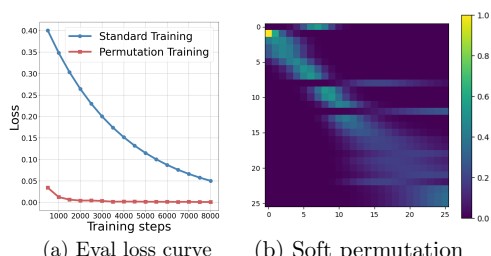

(a) Eval loss curve  (b) Soft permutation

Figure 2: (a) Training-loss curves for a vanilla Transformer (blue) and for a model trained with soft-permutation optimization (red). (b) Permutation matrix learned during permutation training. Sparse off-diagonal weights clustered around the main diagonal indicate leakage from future tokens.

## 3 UNRAVELING THE CHAIN OF THOUGHT

Let $S_L$ be the symmetric group of order $L$, i.e., the set of all permutations over $\{1, \ldots, L\}$. We address the problem of discovering a permutation $\pi \in S_L$ over the token sequence (of length $L$) to the Transformer decoder that improves the overall learning effectiveness of the Transformer.

The Transformer decoder generates output sequences in an auto-regressive manner. It is widely known—especially in the context of chain-of-thought prompting—that the order of generation can have a crucial impact on the reasoning ability of Transformers. For example, Figure 1 shows that, in the task of multiplying two integers, the digits of the target integer (each treated as a token) should be presented in reverse order—from lower to higher digits—because this allows the Transformer to compute carries step by step.

More generally, for example, let $X = [x_1, \ldots, x_L]$ be a sequence of numbers, which is the input sequence to the Transformer. If the target sequence is defined by a map $f(x, y)$ that is non-injective with respect to $y$ (e.g., $f(x, y) = \max\{x + y, 0\}$) as $Y = [y_1, \ldots, y_L]$ with $y_1 = x_1$ and $y_{i+1} = f(x_i + y_i)$ for $i > 1$, learning from reverse order $Y^{\mathrm{r}} = [y_L, \ldots, y_1]$ is significantly harder than that from the forward order because of non-injective $f(x, y)$.

We now introduce our formal problem setup and its challenges.

**Formulation.** Let $\Sigma$ be the set of all tokens. We denote the set of all finite token sequences by $\Sigma^*$ and its restriction to length-$L$ sequences by $\Sigma^L$. Let $\mathcal{T}_\theta : \Sigma^* \times \Sigma^L \to \Sigma^L$ be a Transformer with parameter $\theta$. Hereinafter, we assume that the target sequence length is fixed. Now, let $(X, Y) \sim \mathcal{D}$ be an input–target sequence pair $(X, Y)$ with $|Y| = L$ from a joint distribution $\mathcal{D}$. The empirical risk minimizer $\theta_{\mathrm{ERM}}$ with finite sample set $D = \{(X_i, Y_i)\}_{i=1}^m$ and permutation $\pi \in S_L$ is

$$\theta_{\mathrm{ERM}}^\pi = \arg\min_\theta \frac{1}{m} \sum_{i=1}^m \ell(\mathcal{T}_\theta, X_i, \pi(Y_i)), \tag{3.1}$$

with $\ell$ denotes a loss function. Our goal is to discover a permutation $\pi$ that minimizes the expected risk:

$$\pi^* = \arg\min_{\pi \in S_L} \mathbb{E}_{(X,Y)\sim\mathcal{D}} \big[\ell\big(\mathcal{T}_{\theta_{\mathrm{ERM}}^\pi}, X, \pi(Y)\big)\big]. \tag{3.2}$$

A permutation $\pi(Y)$ of a target sequence $Y = [y_1, \ldots, y_L] \in \Sigma^L$ can be represented as a matrix product $YP$, where $P \in \{0, 1\}^{L \times L}$ is a permutation matrix.

**Challenges.** The optimization over permutations is challenging because one has to test all possible permutations, which is $L!$ for those over $\{1, ..., L\}$. One may introduce a soft permutation matrix

$\tilde{P} \in [0,1]^{L \times L}$ and perform empirical risk minimization jointly over $\theta$ and $\tilde{P}$; namely,

$$\min_{\theta, \tilde{P}} \ \frac{1}{m} \sum_{i=1}^{m} \ell(\mathcal{T}_\theta, X_i, Y_i \tilde{P}). \tag{3.3}$$

However, as shown in Figure 2, such an approach leads to an immediate loss drop at the early stage of training, because the soft permutation $\tilde{P}$ causes information leakage from future tokens; each token in $Y_i P$ is a soft mixture of all the tokens in $Y$, which undermines the next-token prediction. Introducing an additional loss that strongly penalizes non-dominant entries in $\tilde{P}$ and encourages it to approximate a hard permutation matrix $P$ can mitigate such leakage. However, training over nearly hard permutation matrices induces a highly non-convex loss surface, and the optimization process is prone to getting trapped in local minima (Mena et al., 2018; Jang et al., 2017).

## 4 PROPOSED METHOD

We introduce our strategy for discovering a suitable permutation of target token sequences. The key idea is to leverage a characteristic of the training dynamics of deep neural networks: they tend to learn easy samples in the early stages of training, and gradually adapt to harder samples later. This phenomenon has been reported in several contexts in the literature, such as Arpit et al. (2017) for learning with noisy labels and Baldock et al. (2021) for identifying difficult examples.

We discovered this is also the case with the training with different decoder token orders, see Figure 3. Exploiting this observation, we proposed to train a Transformer only for a few epochs on a dataset with various orders in mixture and identify learning-friendly orders as "easy samples," for which the loss drops faster.

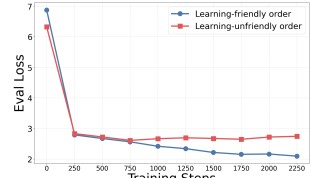

More formally, let $D = \{(X_i, Y_i)\}_{i=1}^{m}$ and $D' = \{(X_i', Y_i')\}_{i=1}^{m'}$ be training and validation sets, respectively. Let $\mathcal{P} = \{P_1, \ldots, P_T\}$ be the set of $T$ candidate permutation matrices. Let $D^{P_t}$ be the set $D$ with reordered target sequences by $P_t$, i.e., $D^{P_t} = \{(X_i, Y_i P_t)\}_{i=1}^{m}$.

Figure 3: Evaluation loss curves when trained with two different orders.

We determine learning-friendly orders through the following *loss profiling*.

**P1.** Let $E \in \mathbb{N}$. Train a Transformer for $E$ epochs on a mixed dataset $\bar{D} := \bigcup_{t=1}^{T} D^{P_t}$. Let $\mathcal{T}_{\theta'}$ be the Transformer after training.

**P2.** Compute the loss on the validation set $D'$ for each permutation; namely, for $t = 1, \ldots, T$, compute

$$\mathcal{L}(D', P_t) = \frac{1}{m'} \sum_{i=1}^{m'} \ell(\mathcal{T}_{\theta'}, X_i', Y_i' P_t). \tag{4.1}$$

Then, the most learning-friendly order $P^* := P_\tau$ is determined with $\tau = \arg\min_t \mathcal{L}(D', P_t)$.

Our experiments empirically observed that a few thousand permutations can be handled at once through this approach. However, the number of permutations grows factorially, which leads us to introduce the following two-stage hierarchical optimization, where aforementioned loss profiling (i.e., **P1** and **P2**) is performed to determine learning-friendly orders at each level.

Figure 4 illustrates our hierarchical method. We start with the initial set of permutation candidates $\mathcal{P}_0 = \{P_1, \ldots, P_T\}$. The global stage splits each token sequence into several blocks and finds a good permutation at the block level. The local stage refines this coarse ordering by permuting the tokens within each block discovered at the global stage. Formally, the two stages operate as follows.

**Global stage.** Let the search depth be $K$ and $T = (K+1)!$. Let $\mathcal{P}_1 := \mathcal{P}_0$. For $k = 1, \ldots, K$, we conceptually split each target sequence into $k$ blocks, [1] and apply the loss profiling to the new

---

[1]When $k = 1$, the sequence is not split into blocks

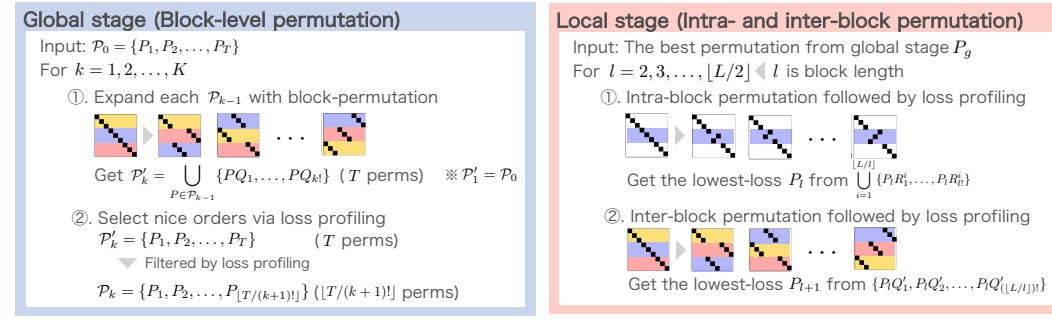

Figure 4: Search flow of our hierarchical approach. **Global stage**: The proposed method generates $T$ candidate permutations by swapping the sequence at the macro-level, exchanging $P$ token blocks to quickly spot coarse, learning-friendly orders. **Local stage**: inside each chosen block, the proposed method further permutes the tokens, refining the sequence to discover a final permutation that maximizes learning ease.

permutation set:

$$\bigcup_{P \in \mathcal{P}_k} \{PQ_1, \ldots, PQ_{k!}\}, \tag{4.2}$$

where $Q_i \in [0,1]^{L \times L}$ are the block-level permutations. The best $\lfloor T/(k+1)! \rfloor$ permutations define the new candidate set $\mathcal{P}_{k+1}$.

We then apply the loss profiling to the final candidate set $\mathcal{P}_{\mathrm{g}} := \mathcal{P}_{K+1}$ and determine the best permutation $P_{\mathrm{g}}$. This permutation is then refined with the local stage.

**Local stage.** Let $P_1 \in P_{\mathrm{g}}$ be the initial permutation. We again conceptually split each target sequence into blocks of size $l$. Let $R_1^i, \ldots, R_{l!}^i \in [0,1]^{L \times L}$ be all the permutations inside the $i$-th block. These permutations do not change the orders outside the $i$-th block. For each block length $l = \{2, 3, \ldots, \lfloor L/2 \rfloor\}^2$, we apply the loss profiling to the new candidate set:

$$\bigcup_{i=1}^{\lceil L/l \rceil} \{PR_1^i, \ldots, PR_{l!}^i\}, \tag{4.3}$$

and denote the lowest-loss result by $P_l$. Keeping each block's internal order fixed, we perform loss profiling over the $\lfloor L/l \rfloor$ block-reordering candidates:

$$\{P_l Q_1', P_l Q_2', \ldots, P_l Q_{\lfloor L/l \rfloor}'\}. \tag{4.4}$$

The best candidate becomes the initial permutation for the next block size $l + 1$.

**Computational overheads.** While the proposed framework repeats training to narrow down the permutations to learning-friendly ones, several aspects keep it practically efficient. First, each of the training runs only for 800–1,600 steps (equivalently, 1–2 epochs with $10^5$ samples of batch size 128) as the difference of loss drop speeds between candidate permutations becomes readily evident in the early stage. Second, a single training can handle a few thousand permutations (up to $7! = 5{,}040$ in our experiments). Third, our global–local framework provides efficient exploration. Specifically, with a global-stage depth of $K$, it needs $K$ runs of training, and the local stage needs $2(\lfloor L/2 \rfloor - 1)$ runs. In our experiments, the longest exploration took 1–7 hours on a single GPU of the NVIDIA A6000ada to find the learning-friendly permutation. It is also worth noting that using a small Transformer model in the exploration is sufficient, as the learning-friendly orders must be universal.

---

[2]When $l$ does not divide $L$, the remaining $L \bmod l$ tokens are placed in an additional block.

# 5 EXPERIMENTS

## 5.1 ORDER-SENSITIVE TASKS

To evaluate the proposed method, we introduce three tasks. They can be learned relatively easily with the forward order, which however becomes challenging with the reverse or random orders.

Let $X = [x_1, x_2, ...]$ be an input sequence and $Y = [y_1, ..., y_L]$ an target sequence of fixed length $L$. At a high level, the target sequence of the three tasks is defined by the following recurrence.

$$y_i = f(X, y_1, \ldots, y_{i-1}), \tag{5.1}$$

where $f(\,\cdot\,)$ is a *non-injective* function with respect to $y_1, \ldots, y_{i-1}$. Namely, other than the forward order, one cannot uniquely determine preceding target tokens $y_1, \ldots, y_{i-1}$ from $X$ and $y_i, \ldots, y_L$. Any disruption of the natural left-to-right order—such as reversing or randomly permuting the targets—breaks the causal chain and substantially increases the learning difficulty.

**RELU.** The recurrence performs the rectified sum:

$$y_1 = x_1 \quad \text{and} \quad y_i = \mathrm{ReLU}\big(x_i + y_{i-1}\big), \quad i = 2, \ldots, L, \tag{5.2}$$

where $\mathrm{ReLU}(z) = \max(z, 0)$. The forward order is trivial to learn because each step depends only on the current input token $x_i$ and the immediately preceding output $y_{i-1}$; in the reverse order, that dependency becomes latent.

**SQUARE-19.** The recurrence performs the squared sum modulo 19 of the $i$-th input token $x_i$ and the previous output token $y_i$ values:

$$y_1 = x_1 \quad \text{and} \quad y_i = x_i^2 + y_{i-1}^2 \mod 19 \in \{-9, \ldots, 9\}, \quad i = 2, \ldots, L. \tag{5.3}$$

The squaring operation is non-injective. The values range in $\{-9, \ldots, 9\}$, and for any $z \in \{-9, \ldots, 9\} \setminus \{0\}$, the preimage of $z^2$ cannot be uniquely determined.

**INDEX.** The recurrence performs input-element pointing based on the latest target tokens.

$$y_1 = x_1 \quad \text{and} \quad y_i = x_p, \quad p = \sum_{j=1}^{d} y_{i-j} \mod L, \quad i = 2, \ldots, L, \tag{5.4}$$

where $d \leq L$ is a fixed window size. Forward order enables the model to compute $p$ incrementally, whereas a reversed or random order destroys the causal chain.

**Example 5.1** (SQUARE-19). Given the input sequence $X = [7, -2, 4, 1, 3]$ and the initial value $y_1 = x_1 = 7$, applying the recurrence in (5.3) produces

$$y_1 = 7, \quad y_2 = (-2)^2 + 7^2 \mod 19 = -4, \quad y_3 = 4^2 + (-4)^2 \mod 19 = -6,$$
$$y_4 = 1^2 + (-6)^2 \mod 19 = -1, \quad y_5 = 3^2 + (-1)^2 \mod 19 = -9.$$

In the forward order, memorizing just $19^2 = 361$ cases suffices to output the target sequence. In reverse order $Y^{\mathrm{r}}$, however, even with $y_5 = -9$ known, $y_4$ is still ambiguous between 1 and $-1$, so learning becomes much harder. Generation examples for the remaining tasks are provided in Appendix D.

Our experiments will focus on the aforementioned three tasks, but the following PROD task will also be used to show that our method can reproduce the observation in Shen et al. (2023).

**PROD.** Given two zero-padded input numbers $a$ and $b$, the target sequence is their product $Y = [ab]$. When the digits are emitted from least significant to most significant, we denote the sequence by $Y$ (forward order); when the digits are emitted in the opposite direction, we denote it by $Y^{\mathrm{r}}$ (reverse order). Unlike the three tasks proposed above, this multiplication task has been examined in earlier studies. Although it does not satisfy the recurrence in (5.1), it still exhibits moderate order sensitivity.

## 5.2 EXPERIMENTAL SETUP

**Datasets.** We generated datasets for the tasks given in Section 5.1. The target length $L$ ranges between $\{5, 6, \ldots, 100\}$. The INDEX task introduces a window size $d \in \{2, 4, 8\}$. The training set

contains 100,000 samples, and the evaluation set contains 1,000 samples. Different random seeds (42 for training and 123 for evaluation) make the two sets disjoint.

**Training setup.** We use the GPT-2 architecture (Radford et al., 2019) of two sizes, small and large. The small model consists of one layer and one attention head, and is used for exploration with the proposed method, while the large one has six layers and is used for the final training with the discovered learning-friendly order to pursue the accuracy. The other parameters are as follows: the embedding and feed-forward dimensions are $(d_{\text{emb}}, d_{\text{ffn}}) = (512, 2048)$, and dropout is set to 0.1. Positional embeddings are randomly initialized and optimized throughout training. The model is trained for 1 and 10 epochs for small and large models, respectively, with AdamW (Loshchilov & Hutter, 2019) ($\beta_1 = 0.9, \beta_2 = 0.999$), a linearly decaying learning rate starting from $5.0 \times 10^{-5}$, and a batch size of 128.

**Exploration setup.** The proposed method uses loss profiling in the global–local pipeline. In the loss profiling, a Transformer is trained on the train set, and then the permutations (i.e., orders) are ranked by the evaluation loss on the evaluation set. The success rate is the proportion of completely correct target sequences generated by the Transformer to the total number of samples in the evaluation set.

**Initialization.** We tested several initializations of the initial candidate set $\mathcal{P}_0$ in the loss profiling (Section 5.4) and global stage (Section 5.5).

- $\mathcal{P}_{\text{g}}$ consists of the identity permutation plus random permutations. For example, if the set size is 100, it includes one identity permutation and 99 random ones.

- $\mathcal{P}_{\text{f}}$ consists of permutations obtained by splitting the forward and reverse orders into column-wise blocks and swapping those blocks.

- $\mathcal{P}_{\text{r}}$ consists of permutations chosen uniformly at random.

- $\mathcal{P}_{\text{b}}$ consists of permutations formed by splitting the forward and reverse sequence into length $b$ and permutes those blocks, and fix $b = 5$ in experiments.

The original target sequences in our dataset are all in the forward order, corresponding to the identity permutation. Examples of these permutation sets are provided in Appendix E.

## 5.3 LEARNING WITH FORWARD AND REVERSE ORDERS

We first show that the learning is easy with forward order, while it becomes significantly challenging with the reverse order. Table 1 reports the success rate when trained with the forward and the reverse orders. As explained in Section 5.1, every task is configured to be learning-friendly in the forward order but learning-unfriendly in the reverse order. Consistent with this design, Table 1 shows that the model almost fully learns each task in the forward order, whereas in the reverse order, the success rate never exceeds roughly 10 %. A closer look at task-specific trends reveals that the success rate for the RELU and SQUARE-19 tasks remains almost unchanged as the target length grows. By contrast, for INDEX, the forward order success rate declines with the window size $d$, indicating that the model struggles when each prediction depends on a larger number of previous outputs.

Table 1: Success rates for the forward and the reverse. The forward order is significantly more learning friendly.

| Task | Target length | Success rate (%) | |
|---|---|---|---|
| | | **Forward** | **Reverse** |
| RELU | $L = 20$ | **99.6** | 0.6 |
| | $L = 50$ | **99.9** | 5.6 |
| | $L = 100$ | **99.4** | 0.0 |
| SQUARE-19 | $L = 20$ | **100** | 0.1 |
| | $L = 50$ | **100** | 0.0 |
| | $L = 100$ | **100** | 0.0 |
| INDEX | $L = 13, d = 2$ | **100** | 9.8 |
| | $L = 13, d = 4$ | **62.3** | 1.3 |
| | $L = 13, d = 8$ | **81.8** | 2.2 |
| | $L = 31, d = 2$ | **100** | 0.8 |

## 5.4 LOSS PROFILING FOR DISCOVERING THE FORWARD ORDER

We next justify the loss profiling before the proposed global–local pipeline. We trained a Transformer on a set of permutations, $\mathcal{P}_{\text{g}}$, containing one learning-friendly forward order (ID=0) and 127 randomly generated learning-unfriendly orders. We set $L = 50$ for the RELU and SQUARE-19 tasks. For the INDEX task, we set $L = 31$ and $d = 4$.

Figure 5(a) shows the evaluation loss for each of the 128 permutations at loss profiling. The forward order (ID=0) achieves the lowest loss among other orders across all tasks, suggesting one can select

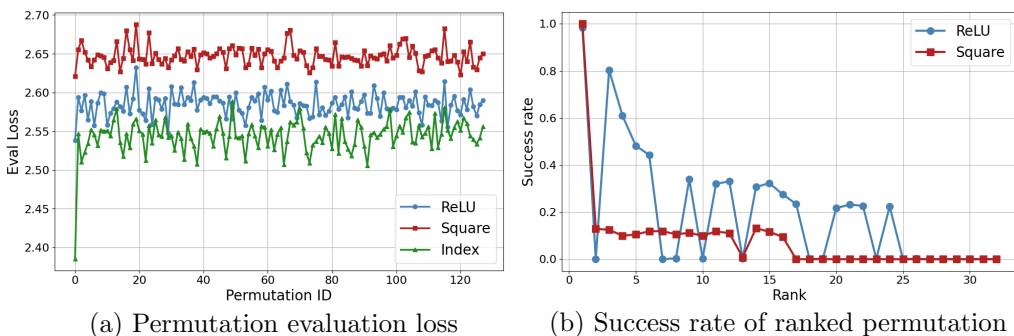

(a) Permutation evaluation loss    (b) Success rate of ranked permutation

Figure 5: (a) Evaluation loss for each permutation obtained via loss profiling. ID=0 corresponds to the forward order, while the others are randomly generated permutations. (b) Success rate of the model when retrained on permutations ranked by the loss values from (a). Permutations are ordered on the x-axis from best (left) to worst (right).

Table 2: The orders discovered by the proposed method in its global and local stages. Depth denotes the hierarchy level $K$ reached in the global stage. Each order is listed relative to the forward sequence; when the list starts at 0, the forward order has been recovered. Forward orders identified at a given stage are highlighted in bold.

| Task | Target Length | Depth | Order after global stage | Discovered final order |
|------|--------------|-------|--------------------------|------------------------|
| RELU | $L = 7$ | $K = 4$ | $[6, 0, 5, 2, 3, 4, 1]$ | $[2, 3, 4, 5, 0, 6, 1]$ |
| | $L = 8$ | $K = 4$ | $[0, 2, 1, 3, 4, 5, 6, 7]$ | $[\mathbf{0, 1, 2, 3, 4, 5, 6, 7}]$ |
| | $L = 9$ | $K = 5$ | $[\mathbf{0, 1, 2, 3, 4, 5, 6, 7, 8}]$ | $[\mathbf{0, 1, 2, 3, 4, 5, 6, 7, 8}]$ |
| | $L = 10$ | $K = 6$ | $[6, 7, 8, 9, 5, 4, 2, 3, 1, 0]$ | $[4, 5, 6, 7, 8, 9, 0, 1, 2, 3]$ |
| | $L = 11$ | $K = 6$ | $[8, 9, 10, 7, 6, 5, 4, 3, 2, 1, 0]$ | $[\mathbf{0, 1, 2, 3, 4, 5, 6, 7, 8, 9, 10}]$ |
| | $L = 12$ | $K = 6$ | $[6, 7, 8, 9, 10, 11, 5, 4, 2, 3, 1, 0]$ | $[1, 2, 3, 4, 0, 5, 6, 7, 8, 9, 10, 11]$ |
| | $L = 13$ | $K = 6$ | $[11, 12, 10, 9, 8, 7, 6, 5, 4, 2, 3, 1, 0]$ | $[\mathbf{0, 1, 2, 3, 4, 5, 6, 7, 8, 9, 10, 11, 12}]$ |
| SQUARE-19 | $L = 7$ | $K = 4$ | $[\mathbf{0, 1, 2, 3, 4, 5, 6}]$ | $[\mathbf{0, 1, 2, 3, 4, 5, 6}]$ |
| | $L = 8$ | $K = 4$ | $[1, 2, 4, 5, 0, 6, 7, 3]$ | $[1, 2, 4, 5, 0, 6, 7, 3]$ |
| | $L = 9$ | $K = 5$ | $[\mathbf{0, 1, 2, 3, 4, 5, 6, 7, 8}]$ | $[\mathbf{0, 1, 2, 3, 4, 5, 6, 7, 8}]$ |
| | $L = 10$ | $K = 6$ | $[9, 8, 7, 6, 5, 4, 3, 2, 1, 0]$ | $[\mathbf{0, 1, 2, 3, 4, 5, 6, 7, 8, 9}]$ |
| | $L = 11$ | $K = 6$ | $[\mathbf{0, 1, 2, 3, 4, 5, 6, 7, 8, 9, 10}]$ | $[\mathbf{0, 1, 2, 3, 4, 5, 6, 7, 8, 9, 10}]$ |
| | $L = 12$ | $K = 6$ | $[1, 2, 3, 4, 5, 6, 7, 11, 10, 9, 0, 8]$ | $[\mathbf{0, 1, 2, 3, 4, 5, 6, 7, 8, 9, 10, 11}]$ |
| | $L = 13$ | $K = 6$ | $[0, 1, 2, 3, 12, 11, 10, 4, 5, 6, 7, 8, 9]$ | $[8, 9, 0, 1, 2, 3, 4, 10, 11, 12, 5, 6, 7]$ |
| INDEX | $L = 13, d = 2$ | $K = 6$ | $[1, 0, 2, 3, 4, 5, 6, 7, 8, 9, 10, 11, 12]$ | $[\mathbf{0, 1, 2, 3, 4, 5, 6, 7, 8, 9, 10, 11, 12}]$ |
| | $L = 13, d = 4$ | $K = 6$ | $[0, 1, 7, 6, 4, 2, 5, 8, 3, 9, 10, 11, 12]$ | $[0, 1, 7, 6, 4, 2, 5, 8, 3, 9, 10, 11, 12]$ |
| | $L = 13, d = 8$ | $K = 6$ | $[1, 2, 3, 4, 5, 6, 7, 8, 10, 9, 12, 0, 11]$ | $[1, 2, 3, 4, 5, 6, 7, 8, 10, 9, 12, 0, 11]$ |
| PROD | $L = 10$ | $K = 6$ | $[\mathbf{0, 1, 2, 3, 4, 5, 6, 7, 8, 9}]$ | $[\mathbf{0, 1, 2, 3, 4, 5, 6, 7, 8, 9}]$ |

a learning-friendly permutation via loss profiling if the set contains one. This effect is particularly pronounced in the INDEX task, which is the hardest task among the three.

Now, we ranked the 128 orders according to the evaluation losses shown in Figure 5 and then trained a Transformer for each of the top 32 orders. Figure 5(b) shows that the success rate generally aligns with the rank; training with a highly ranked order leads to a high success rate in the RELU and SQUARE-19 tasks.

For the INDEX task, which is the hardest task among the three, the success rate was all close to zero (omitted from the plot). Still, the top-ranked order (i.e., forward order) is the most learning-friendly order by the construction of the task. This result indicates that the loss profiling is more advantageous in finding *implicit* learning-friendly orders than exhaustively repeating full training and success rate evaluation, even ignoring the computational burden of the latter. The result also justifies using small Transformers in the exploration stage, even for hard tasks. One only needs to use large and powerful models in the final training with the discovered order.

### 5.5 GLOBAL–LOCAL METHOD WITH LOSS PROFILING

We now demonstrate that the proposed method can discover the learning-friendly orders up to $L = 13$ (i.e., roughly 6 billion possible orders) with random initialization $\mathcal{P}_r$ and $L = 40$ with a structured

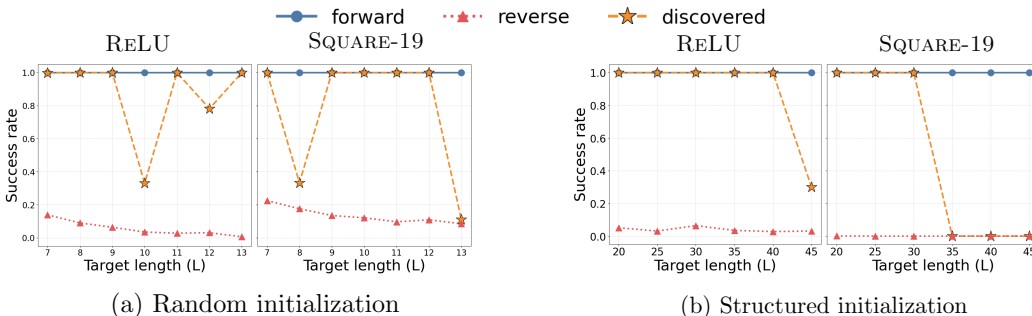

(a) Random initialization          (b) Structured initialization

Figure 6: Success rates of orders found by our hierarchical search using different search-space initializations. (a) Search initialized with a fully random permutation set $\mathcal{P}_{\mathrm{r}}$. (b) Search initialized with a structured, block-restricted set $\mathcal{P}_{\mathrm{b}}$. The colors represent the forward (blue), reverse (red), and our discovered (yellow) orders.

initialization $\mathcal{P}_{\mathrm{b}}$. One should use the former if no prior knowledge of the ordering is available, while the latter (or something similar) can be designed for some tasks. For example, on polynomial tasks, one may permute the tokens at a monomial level, which naturally leads to initialization with block-level permutations.

**Initialization with $\mathcal{P}_{\mathrm{r}}$.** Table 2 shows the permutation discovered at the global stage and the final one. After the global stage, tokens that are neighbors in the input usually remain adjacent, showing that the method first captures coarse structure. The subsequent local stage then fine-tunes this order and moves the order closer to the optimal forward arrangement. For the RELU and SQUARE-19 tasks, global orders were often already learning-friendly, and retraining a model on them always produces a higher success rate than training on the reverse order (see Figure 6(a)). The INDEX task proves harder: as the reference width $d$ grows, learning is difficult even in the forward order (see Section 5.1), which flattens the loss landscape and makes good permutations harder to rank. In the PROD, the proposed method succeeds in rediscovering the least-significant-digit first order reported by Shen et al. (2023), and it finds the optimal order for target lengths up to 13, identifying a single solution among roughly $13! \approx 6 \times 10^9$ possibilities.

**Structured initialization with $\mathcal{P}_{\mathrm{b}}$.** When the search is initialized with $\mathcal{P}_{\mathrm{b}}$, the proposed method scales to much longer target sequences. Figure 6(b) shows the resulting success rate curves for RELU and SQUARE-19: the optimal order is found for both tasks up to $L = 30$, and for RELU even at $L = 40$. At $L = 40$, the theoretical permutation space still contains about $10^{47}$ elements, indicating that once implausible candidates are pruned, the proposed method can explore the remaining space far more effectively. These results demonstrate that our hierarchical search can recover optimal orders in both the most challenging fully random scenario and the more realistic, block-restricted setting, and that its advantage grows as the candidate space is made more coherent.

## 6 CONCLUSION

This study addressed a new task of reordering decoder input tokens for Transformers in learning arithmetic tasks. In essence, the proposed method performs short-term training on a mixture of target sequences in different orders and discovers easy samples for which the loss drops faster, as learning-friendly orders. To search the factorially large space efficiently, we propose a two-stage hierarchical approach combining global block-level exploration with local refinement. The experiments on three order-sensitive arithmetic tasks (RELU, SQUARE-19, and INDEX) demonstrated that the proposed method discovers a learning-friendly order, improving the success rate from about 10 % to near 100 % and works for target lengths up to 13 tokens ($13! > 6 \times 10^9$ permutations). Moreover, it rediscovered the reverse-digit order reported in earlier work on the PROD task. This study presents an automatically unraveling chain of thought that markedly enhances a Transformer's reasoning ability. The extension to longer sequences and target sequences at a variable length will be future work.

**Reproducibility Statement and the Use of LLMs.** Common details of our experimental setup are described in Section 5.2, while any parameters or settings specific to each experiment are provided at the beginning of each experiment in Section 5. The source code used for experiments is provided as supplemental material and will be publicly available after clean-up.

The LLMs were used for assistance purposes only. We used them to improve our writing, speed up coding, and to assist in part of our literature search by listing papers with OpenAI's ChatGPT (OpenAI, 2024). No essential contributions were made by the LLMs.

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

## A  VISUALIZING ATTENTION MAP

We present the attention maps obtained when training a Transformer on our proposed RELU task with datasets reordered in different ways. For this analysis, we use a GPT-2 model with a single layer and a single attention head. Figure 7 shows the attention maps for target length $L = 20$ under four target orders. The forward and reverse orders are defined in Section 5.1. The one-permuted order swaps exactly one pair of adjacent target tokens, whereas the random order is a random permutation of the forward sequence. Figure 8 illustrates how the attention maps change as the target length increases.

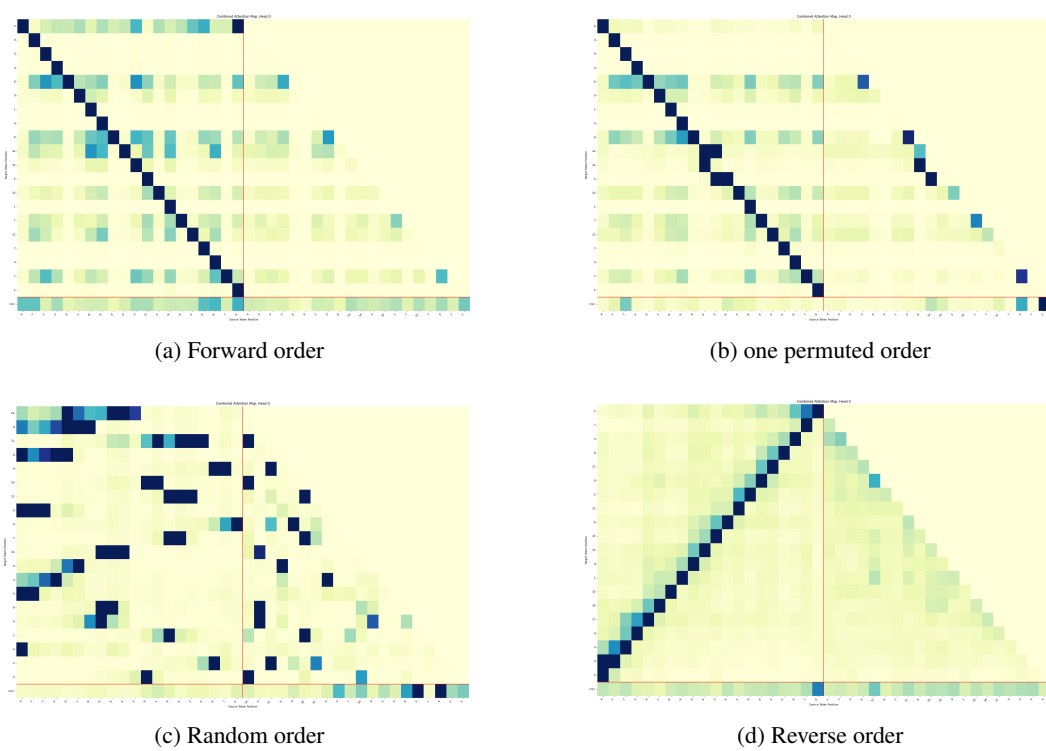

(a) Forward order

(b) one permuted order

(c) Random order

(d) Reverse order

Figure 7: Attention matrices from models trained with four different target orders in the RELU task.

Table 3: Attention sparsity $S$ across target orders. A smaller value of $S$ indicates greater sparsity.

| Task | Target length | Sparsity | |
|---|---|---|---|
| | | **Forward** | **Reverse** |
| RELU | $L = 20$ | **1.160** | 1.640 |
| | $L = 50$ | **1.462** | 4.319 |
| | $L = 100$ | **1.687** | 3.195 |
| SQUARE-19 | $L = 20$ | **1.117** | 1.531 |
| | $L = 50$ | **1.773** | 1.914 |
| | $L = 100$ | **1.407** | 1.990 |
| INDEX | $L = 13, d = 2$ | **0.848** | 2.574 |
| | $L = 13, d = 4$ | **0.887** | 1.486 |
| | $L = 13, d = 8$ | **1.116** | 1.596 |

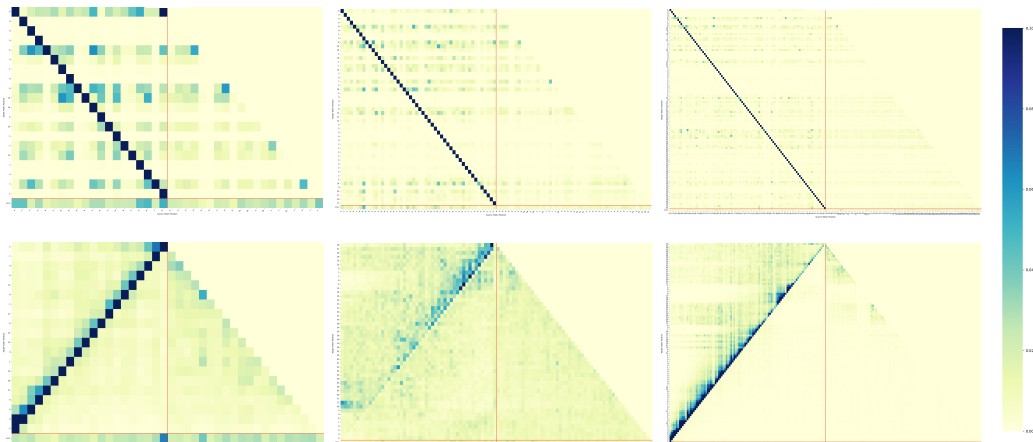

Figure 8: Differences in the attention matrices for the RELU task between forward and reverse orderings. The top three matrices correspond to models trained with forward order, and the bottom three with reverse order. Each pair of matrices shows results for input lengths $n = 20, 50$, and $100$, respectively.

## B  SOFT-PERMUTATION OPTIMIZATION VIA ATTENTION SPARSITY

**Analysis of attention sparsity.** To address the challenges of permutation optimization, we undertake a more detailed analysis. Intuitively, when the target order is learning-friendly, the causal structure of the sequence is broken: more input and output tokens become relevant to predicting the next token. Conversely, for an learning-friendly order we expect the attention map to be *sparser*.

Let the query and key matrices be $Q, K \in \mathbb{R}^{L' \times d_{\text{emb}}}$, where $L'$ is the decoder-input length and $d_{\text{emb}}$ the embedding dimension. The self-attention weights are

$$A = \text{Softmax}\left(\frac{QK^\top}{\sqrt{d_{\text{emb}}}}\right) \in \mathbb{R}^{L' \times L'}, \tag{B.1}$$

where $\text{Softmax}(\cdot)$ is applied row-wise. Because each row of $A = (a_{ij})_{ij}$ is a probability vector, we define the mean sparsity $S$ by the Shannon entropy:

$$S = -\frac{1}{L'} \sum_{i,j=1}^{L'} a_{ij} \log a_{ij}. \tag{B.2}$$

We compute $S$ for models trained on both the forward (learning-friendly) and reverse (learning-unfriendly) orders of the order-sensitive tasks (section 5.1). table 3 shows that the forward order consistently yields lower $S$, and—since a smaller $S$ directly means higher sparsity—this confirms that learning-friendly orders produce sparser attention. Representative heat maps are provided in appendix A.

Because $S$ is derived from the learned attention weights, it is independent of the language-model loss and can serve as an orthogonal diagnostic metric. We also experimented with optimizing permutations under an additional sparsity regularizer that rewards low-entropy attention (cf. appendix B). Even with this bias, the optimizer failed to discover the learning-friendly order and instead converged to interleaved permutations, suggesting that sparsity alone is insufficient to solve the permutation search in difficult regimes.

We present a soft-permutation optimization method based on attention sparsity. In our two-stage strategy, we first optimize the Transformer parameters $\theta$ by minimizing the standard sequence-modeling loss over the training set:

$$\min_\theta \frac{1}{m} \sum_{i=1}^{m} \ell\left(\mathcal{T}_\theta, X_i, Y_i\right). \tag{B.3}$$

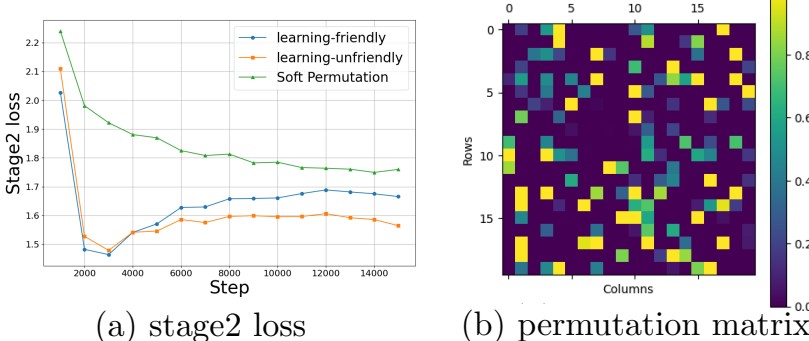

(a) stage2 loss        (b) permutation matrix

Figure 9: (a) Comparison of Stage 2 loss under fixed learning-friendly order, fixed learning-unfriendly order, and learned soft permutation. (b) shows a visualization of the learned soft permutation.

Next, denoting by $A = [a_{ij}] \in \mathbb{R}^{L' \times L'}$ the attention map produced when the target sequence is fed as $Y_i \tilde{P}$ into the Transformer, we optimize the soft permutation $\tilde{P}$ by minimizing the total attention entropy:

$$\min_{\tilde{P}} \frac{1}{L'} \sum_{i=1}^{L'} \sum_{j=1}^{L'} a_{ij}. \tag{B.4}$$

In the experiments, we alternate between the two-stage optimizations at each step. Figure 9 compares the stage 2 loss under three conditions: the fixed, learning-friendly order, the fixed, learning-unfriendly (reverse) order, and the learned soft permutation. We observe that the soft permutation does not reduce the entropy-based loss (B.4) relative to the fixed orders, nor does it yield a genuinely hard ordering. Because attention sparsity—as measured by total attention mass—decreases even for static orders, it cannot serve as a reliable objective for permutation optimization.

## C    Permutation Search via Evolutionary Strategy

This section summarizes the evolutionary strategy (ES) baseline that we ran in parallel with our proposed method to search the permutation space. Each individual is a permutation $P$; its fitness is the (negative) early-stage training loss of a Transformer trained with that order, so that permutations that are easier to learn receive higher scores. The ES is controlled by the population size $N_p$, crossover probability $N_c$, mutation probability $N_m$, number of generations $N_g$, tournament size $N_t$, and elitism ratio $N_r$, and proceeds as follows:

(1) **Population initialization:** sample $N_p$ random permutations.

(2) **Selection:** pick parents via tournament selection with size $N_t$.

(3) **Crossover:** with probability $N_c$, apply partially–mapped crossover to each selected pair.

(4) **Mutation:** with probability $N_m$, swap two positions in the offspring permutation.

(5) **Elitism:** evaluate every individual by

$$\text{fitness}(P) = -\frac{1}{m'} \sum_{i=1}^{m'} \ell\big(\mathcal{T}_\theta, X_i, Y_i P\big),$$

and copy the top $N_r$ fraction to the next generation.

(6) **Termination:** stop when $N_g$ generations have been processed.

Table 4 lists the permutation identified by the evolution strategy (ES) and the performance obtained when the model is retrained using that order.

Table 4: Success rate obtained when the Transformer is retrained on the permutations discovered by the ES.

| Task | Input length | ES-discovered order | Success rate (%) | |
|---|---|---|---|---|
| | | | **Retrain** | **Reverse** |
| RELU | $L = 5$ | [2, 1, 0, 4, 3] | 26.9 | 10.4 |
| | $L = 10$ | [0, 1, 2, 3, 4, 5, 6, 7, 8, 9] | 100 | 3.5 |
| | $L = 20$ | [6, 7, 9, 8, 12, 11, 13, 18, 17, 14, 16, 15, 19, 5, 10, 1, 0, 3, 4, 2] | 9.2 | 0.7 |
| SQUARE-M19 | $L = 5$ | [1, 2, 3, 4, 0] | 100 | 21.5 |
| | $L = 10$ | [3, 4, 5, 6, 7, 8, 9, 1, 0, 2] | 99.9 | 13.5 |
| | $L = 20$ | [9, 10, 11, 12, 13, 14, 2, 3, 4, 5, 6, 15, 16, 17, 18, 19, 0, 1, 7, 8] | 5.2 | 1.2 |
| INDEX ($m = 2$) | $L = 13$ | [0, 1, 2, 3, 4, 10, 9, 5, 6, 12, 11, 7, 8] | 27.6 | 7.8 |

## D EXAMPLE DATASET

This section provides concrete examples for the four tasks introduced in Section 5.1. Table 5 summarizes the correspondence between the input $X$ and the target $Y$, with every $Y$ given in the forward—that is, learning-friendly—order. For the PROD task only, the input consists of two integers, $a$ and $b$.

Table 5: Representative input–output samples for each task

| Task | Input | Target |
|---|---|---|
| RELU, $L = 50$ | $X = (4, -7, -7, -3, 8, 1, -8, -9, 8, 6$ 
 $0, -9, 5, -9, 6, 5, -5, -9, 7, -5$ 
 $8, -6, -7, -2, -7, 6, 7, -2, 0, -6$ 
 $-3, -8, -7, -8, 3, -1, -6, 1, -4, -9$ 
 $2, -7, 1, 4, 9, -5, 6, 2, 3, -3)$ | $Y = (4, 0, 0, 0, 8, 9, 1, 0, 8, 14$ 
 $14, 5, 10, 1, 7, 12, 7, 0, 7, 2$ 
 $10, 4, 0, 0, 0, 6, 13, 11, 11, 5$ 
 $2, 0, 0, 0, 3, 2, 0, 1, 0, 0$ 
 $2, 0, 1, 5, 14, 9, 15, 17, 20, 17)$ |
| SQUARE-M19, $L = 50$ | $X = (-5, -9, 8, 7, 8, -7, 5, -9, -6, 9$ 
 $-2, -8, 6, -7, 2, -7, -6, -5, -5, 7$ 
 $3, 6, -9, 1, 7, 0, -7, 7, -5, 0$ 
 $-2, 6, -1, -9, -6, -7, 0, 2, 7, -1$ 
 $1, -2, -6, -7, 5, 1, 9, -6, -3, -3)$ | $Y = (-5, 2, 2, 6, -4, -1, -2, 0, 8, 3$ 
 $4, -5, -5, 8, 2, 6, 6, -5, 3, -8$ 
 $7, 0, -4, 8, 9, -4, -1, 3, 6, 8$ 
 $2, -7, 3, 5, -5, 8, -2, -1, 3, 1$ 
 $-7, 6, 6, 0, -3, 1, -3, -2, 4, -3)$ |
| INDEX, $L = 13, d = 2$ | $X = (1, 5, 12, 3, 8,$ 
 $6, 11, 12, 2, 8, 10, 8, 10)$ | $Y = (5, 6, 8, 5, 1, 11,$ 
 $10, 2, 10, 10, 12, 8, 12)$ |
| INDEX, $L = 13, d = 4$ | $X = (1, 5, 12, 3, 8,$ 
 $6, 11, 12, 2, 8, 10, 8, 10)$ | $Y = (5, 6, 8, 5, 1, 11,$ 
 $10, 2, 10, 10, 12, 8, 12)$ |
| INDEX, $L = 13, d = 8$ | $X = (1, 5, 12, 3, 8,$ 
 $6, 11, 12, 2, 8, 10, 8, 10)$ | $Y = (5, 6, 8, 5, 1, 11,$ 
 $10, 2, 10, 10, 12, 8, 12)$ |
| PROD, $L = 10$ | $a = (0, 0, 2, 0, 3)$ 
 $b = (0, 2, 6, 3, 7)$ | $Y = (1, 1, 3, 5, 3, 5, 0, 0, 0, 0)$ |

## E EXAMPLE SET OF PERMUTATIONS

This section describes the four permutation sets introduced in Section 5.2. Figure 10 visualizes every permutation $P$ in those four sets. Each set contains 32 elements.

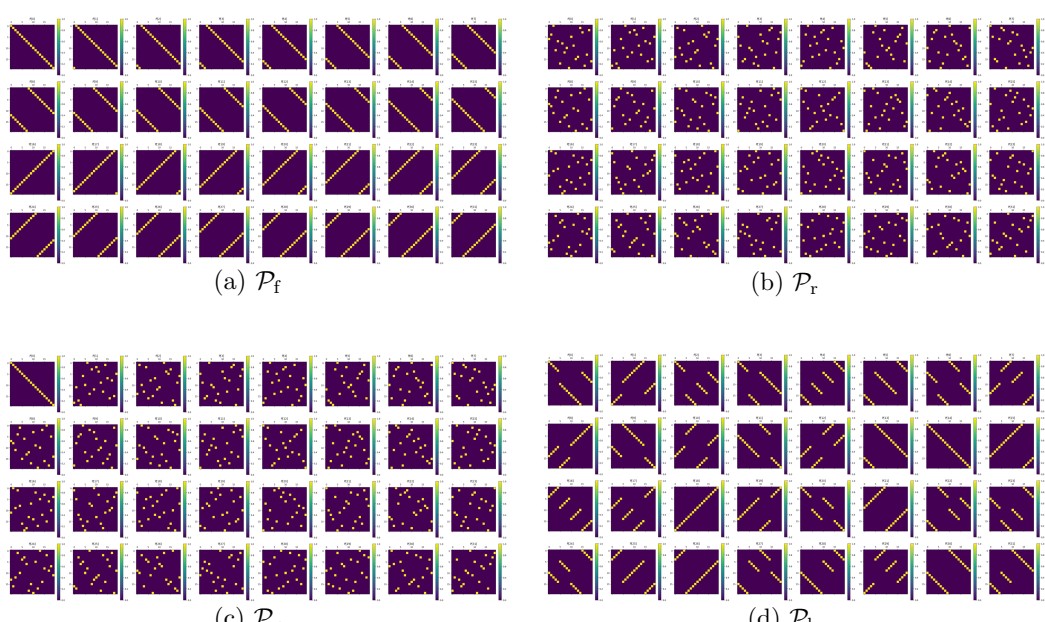

(a) $\mathcal{P}_\text{f}$

(b) $\mathcal{P}_\text{r}$

(c) $\mathcal{P}_\text{g}$

(d) $\mathcal{P}_\text{b}$

Figure 10: Visualization of the elements in the four permutation sets.

