# OpenReview forum: "Chain of Thought in Order: Discovering Learning-Friendly Orders for Arithmetic"
_ICLR.cc/2026/Conference — ICLR 2026 Conference Withdrawn Submission_

### Official Review · Reviewer_1dB3 · 2025-10-23

**Soundness:** 3
**Presentation:** 3
**Contribution:** 2
**Rating:** 4
**Confidence:** 3

**Summary:**

This paper addresses discovering learning-friendly orderings for input tokens when training Transformers on arithmetic tasks. The authors propose training a Transformer on a mixture of target sequences in different orders, then identifying learning-friendly orders as those with fast loss drops in early training. To handle factorial search space growth, they introduce a two-stage hierarchical approach with global block-level reordering followed by local intra-block refinement.

The method is validated on three custom order-sensitive arithmetic tasks, successfully discovering learning-friendly orders from up to ≈ 6 billion candidates and improving success rates from ~10% to 100%. When applied to multiplication, it automatically rediscovered the previously reported reverse-digit order (least-to-most significant).

**Strengths:**

The experiments are clearly explained and executed with the results well presented.

This new approach to prompt optimization can be used to reduce model training effort in applicable use cases - especially cases where the model can’t learn a task without reordering tokens but can with reordered tokens.

The token ordering selection mechanism is novel and seems efficient. It is a necessary prerequisite for the new approach to be feasible at non-toy scale.

**Weaknesses:**

CoT typically refers to LMs explicitly generate intermediate reasoning steps before reaching a final answer, whereas this paper uses it to mean "sequence of generation steps" or "order of tokens" which is a stretch. The paper is about discovering the optimal order in which to arrange output tokens during training. The paper title would be better as just “Discovering Learning-Friendly Orders for Arithmetic”.

As I understand the paper, in the 4 use cases studied, the optimal ordering is reversing the token ordering. The paper would be stronger if the technique showed that learning a (5th) task is optimized by a previously-unknown optimal token ordering (that is just the “reverse” ordering). Without such an example, the technique’s usefulness over other existing techniques is undemonstrated.

Suppose a novel token ordering for a 5th task is found. A model successfully trained with this technique can now do the task but will permanently need specially ordered prompt tokens. This seems to make the technique pretty specialized.

The claim “as the learning-friendly orders must be universal” may well be true but is not shown to be true.

Minor:
- Related Work is short.
- Change cite usage in “(Arpit et al., 2017) has experimentally”

**Questions:**

- What is the evidence that this technique reveal insights not provided by other existing techniques?

- In Table 2, are all or some of the “discovered final orders” learnt optimal in some sense? Or only the bolded ones? Or are they just the best ones found during (limited) training?

- This technique seems very specialized. What is cost trade-off / justification: when should we optimize token ordering to reduce training costs or improve performance?

- Once a model has learn a task via this technique, can the model be further refined to also handle the natural token ordering?

---

### Official Review · Reviewer_tkf6 · 2025-11-02

**Soundness:** 2
**Presentation:** 3
**Contribution:** 2
**Rating:** 4
**Confidence:** 3

**Summary:**

This paper studies a method to automatically discover better output orderings for chain-of-thought reasoning.
The authors evaluate three tasks: ReLU, SQUARE-19, and Index.
The proposed approach has two components: loss profiling and two-stage hierarchical optimization. Loss profiling trains a Transformer on a mixture of permutations and selects the permutation with the lowest validation loss, based on the intuition that a Transformer learns faster when given a good chain-of-thought ordering.
The two-stage hierarchical optimization searches the permutation space and, combined with loss profiling, aims to identify a learning-friendly order from which the model can easily learn the task.
Experiments on the three arithmetic tasks show that the method can discover learning-friendly orders.

**Strengths:**

**S1.**
Automatic chain-of-thought ordering is important: common practice relies on human priors to design an ordering, which may be suboptimal for Transformers.

**S2.**
The paper is well organized and provides clear examples.

**Weaknesses:**

**W1.**
The key claim is that the proposed method can find a good ordering that enables a Transformer to learn efficiently.
To convincingly support this, the paper should demonstrate that it reliably identifies such orderings across tasks that are not inherently forward- or backward-friendly.
As written, ReLU, SQUARE-19, and Index appear forward-friendly, and Prod is backward-friendly.

I would be more convinced if the authors included tasks that are not inherently forward- or backward-friendly.
For instance, fix permutation $\sigma \in [L] \rightarrow [L]$ and define:
(1) $y_{\sigma(1)} = x\_1$.
(2) $y_{\sigma(i)} = ReLU(y_{\sigma(i-1)} + x_{i})$ for $i = 2, \dots, L$.
Such constructions could test whether the method can recover a truly task-specific optimal order rather than relying on an obvious forward/backward bias.

**W2.**
Some important details seem missing (apologies if I missed them):

(1) I don’t see which distribution is used to sample the $x_i$ for each task.

(2) I don’t see how robust your method is.

For example, in Lines 466-467, you mentioned that “the optimal order is found for both tasks up to $L=30$, and for ReLU even at $L=40$”.
However, I don’t see how often is the optimal order found: always, often, or only occasionally?
Reporting rates (and variance across random seeds) would clarify robustness.

**Questions:**

**Q1.**
In figure 5(a), for SQUARE task, the evaluation losses of the optimal and suboptimal permutations are quite close. As $L$ increases and the task becomes harder, loss profiling might discard the optimal permutation. In such cases, can the two-stage optimization rediscover the optimal permutation?

**Q2.**
In Table 1, why is the success rate for $(L,d)=(13,4)$ lower than for $(13,8)$?
Intuitively, a smaller $d$ might seem easier.
Also, how many trials were run to compute the success rates in Table 1?

---

### Official Review · Reviewer_qZyA · 2025-11-02

**Soundness:** 1
**Presentation:** 1
**Contribution:** 2
**Rating:** 2
**Confidence:** 4

**Summary:**

This paper proposes a method to discover a learning-friendly ordering of target sequence tokens in a decoder input to solve an arithmetic task with a Transformer. The learning-friendly ordering of target tokens is determined by loss profiling, training a single model on a mixture of target sequences in different orders and choosing the ‘easiest’ one with respect to validation loss. The authors propose a two-stage heuristic combining global and local search of permutations for the sake of factorially large search space. The method is numerically tested with several arithmetic tasks: RELU (outputting a sequence of clipped cumulative sum), SQAURE-19 (modular arithmetic of sum-of-squares), INDEX (input-element pointing based on a partial sum of latest target tokens), and Prod (zero-padded integer multiplication).

**Strengths:**

1. The paper pinpoints a novel problem that the ordering of target sequence (i.e., answer) tokens is important in learning an arithmetic task with a Transformer.
2. The paper addresses this problem by proposing a learning-based heuristic to discover a learning-friendly ordering of answer tokens.
3. The paper provides an ablation study about the design choice of their hierarchical algorithm so that the two-stage method and the structured initializations can much improve the efficiency in finding a learning-friendly permutation for tasks with a longer target length.

**Weaknesses:**

1. The paper proposes an invalid definition of ‘order-sensitivity’ for a task, defined with a ‘non-injective’ mapping $f(x, y)$ w.r.t. $y$. Indeed, in the RELU, SQUARE-19, and INDEX tasks, the mappings of form $y_i = f_i(X, y_{<i})$ used for defining the tasks are not injective in $y_{<i}$, if we only have access to the $i$-th prompt token $x_i$. However, given that we have an entire prompt $X=[x_1, \cdots, x_L]$, then each $y_i$ is uniquely determined for all $i=1,\cdots,L$.  For example, for the RELU task, we have $y_1=x_1$ and $y_i = \max_{k=1, 3, 4, \ldots, i+1} \sum_{j=k}^{i} x_i$ by definition. Thus, the authors’ claim that the tested synthetic tasks are using non-injective mapping in $y_{<i}$ seems wrong. The paper should have provided a more sophisticated and valid definition of order-sensitivity. For example, one can compute a minimal number of variables among $x_1, \ldots, x_L$, $y_1, \ldots, y_{i-1}$, $y_{i+1}, \ldots, y_L$ to uniquely determine $y_i$, and then check whether such (minimally required) variables are always appearing before $y_i$ in each ordering of answer tokens.
2. The paper is not motivating their two-stage search flow well. First of all, they did not rigorously define several jargons, such as ‘block-permutation’, ‘intra-block permutation’ and ‘inter-block permutation’. Second, the overall readability of Section 4 is quite poor because there are quite a lot typos in the description of the search flow. Third, it is not intuitively clear why alternating the searches over intra-block and inter-block permutation is helpful in terms of factorially large search space. Lastly, the numerical performance of their algorithm is quite questionable (e.g., Table 2) without using a prior knowledge (i.e., structured initialization) about the tasks.
3. The paper should have mentioned that the multiplication task (PROD) with a reversed digit ordering satisfies the recurrence in Equation (5.1). That is, the sentence “Although it does not satisfy the recurrence in (5.1), …” (in line 319) is not true.
4. The term ‘chain-of-thought’ is misused in this paper. As far as I know, it should refer to an intermediate steps of solving the problem, rather than auto-regressive method of generating the answer itself. However, from section 3, the paper is only focusing on the ordering of answer tokens.
5. Learning paradigms other than next-token-prediction (e.g., teacher-less learning [1] or full-output prediction [2]) are worth to be discussed. This is mainly because, although the authors argue that the soft-permutation-based order-finding method is not working in a usual teacher-forced next-token-prediction setup because of the information leakage, this claim may not be true in other learning paradigms.

---

References:

[1] Bachmann et al., The Pitfalls of Next-Token Prediction, ICML 2024

[2] Fan et al., Looped Transformers for Length Generalization, ICLR 2025

**Questions:**

1. I think we can improve the efficiency of permutation-searching algorithm much more because the target length is fixed to be $L$ in this paper. For example, we can train/test $L$ different permutations (which are equivalent up to circulation) in parallel. This can be done by expanding the target length to be $2L-1$ as $(y_1, y_2, \cdots, y_L, y_1, y_2, \cdots, y_{L-1})$, although you need to manipulate the attention mask manually. I believe a similar idea can be applied in several ways, in both global/local stages of the algorithm. If the authors have enough time, can they numerically test this idea whether it can accelerate finding a learning-friendly permutation?
2. How can we apply/extend the proposed method to a task having a variable target length? I believe this is a valid question because most of the arithmetic tasks (e.g., addition, multiplication, …) does so.
3. I wonder whether the authors had a chance to come up with a class of order-sensitive tasks whose learning-friendly order is not straightforward to guess. What will happen if the proposed algorithm is applied to such tasks? I am also looking forward to a further numerical results to answer this question.

---

### Official Review · Reviewer_XWqm · 2025-11-03

**Soundness:** 3
**Presentation:** 3
**Contribution:** 2
**Rating:** 4
**Confidence:** 3

**Summary:**

This paper studies reordering the decoder’s target tokens (the chain-of-thought steps) so Transformers learn arithmetic tasks more easily. It trains on a mixture of candidate orders and chooses the “learning-friendly” ones via early-epoch loss profiles. To reduce the factorial search, it uses a two-stage hierarchical search: 1) first choose block order, 2) reorder within blocks. Across several order-sensitive arithmetic tasks including integer multiplication, the method finds orders that increases success from 10% to 100%, and it also recovers the known reverse-digit order for multiplication.

**Strengths:**

1) It's a very interesting paper framing the order of  intermediate steps as a search over permutations.

2) They shows good performance finding the order to simplify the learning on multiple synthetic tasks specially the method rediscovers the reverse-digit order for multiplication tasks corroborating prior findings.

3) The also proposed a simple hierarchical strategy: block-wise first, then within-block reordering reduces the factorial search cost.

**Weaknesses:**

1) They only evaluated it only  on synthetic arithmetic tasks. How well does the approach transfer to natural-language reasoning e.g. GSM8K, etc.?

2) How sensitive are the selected orders to random seeds, positional encoding and model scale choices? Can you provide variance/error bars across runs?

3) Even short-term training over thousands of permutations can be costly. Can you quantify wall-clock/search cost vs. gains both on Full and hierarchical search.

**Questions:**

Look at the weaknesses.

---

### Author Response · Authors · 2025-12-03
**Global Response: Withdrawal and Appreciation**

Given the unanimous negative scores, we decided to withdraw the current submission.

We understand that because of the recent OpenReview issue, the reviewers will no longer be able to communicate with us. Still, we sincerely thank all reviewers (XWqm, qZyA, tkf6, 1dB3) for their time and insightful, constructive comments, and provide a brief response.


1. **Special thanks to reviewer qZyA**
We would like to extend our special gratitude to reviewer qZyA. Your detailed feedback—particularly regarding the rigorous definition of order-sensitivity (injective mapping) and the critique of our search flow description—was exceptionally constructive. We recognize that our mathematical formulation needs to be more precise, and your specific suggestions provided a clear path to improve the paper's soundness.

2. **Interpretation of Chain of Thought (Response to Reviewers qZyA & 1dB3)**
Regarding the concern that our usage of Chain of Thought (CoT) is a misuse: We respectfully disagree but will make this point clear to avoid any confusion. In arithmetic tasks performed by autoregressive models, the generation of output tokens is the step-by-step reasoning process itself. Therefore, unraveling the optimal order of these tokens effectively means optimizing the reasoning path (CoT) required to reach the solution. While we believe our usage captures the essence of CoT, we acknowledge the potential for confusion and will clarify this conceptual link and refine our definitions in the next version.

3. **Generalization to non-trivial tasks (Response to Reviewers XWqm, tkf6, 1dB3)**
Reviewers questioned whether our method works for tasks where the optimal order is not simply forward or backward (e.g., NLP tasks). While we plan to add such experiments, we argue that our current results serve as a strong proof-of-concept. The fact that our method automatically rediscovered the reverse-digit order for multiplication—without any prior bias—suggests that the underlying search mechanism is capable of identifying learning-friendly structures latent in the data. This implies that for tasks with unknown optimal orders (where human priors fail), our method has the potential to uncover non-trivial, domain-specific ordering.

4. **Robustness and search efficiency (Response to Reviewers XWqm, tkf6)**
Regarding the cost of the hierarchical search: Although we could not include new wall-clock analyses during the rebuttal, we believe the trade-off is highly favorable. In the tasks we analyzed, using a suboptimal order often leads to a failure to learn (success rates near 0%). Compared to the prohibitive cost of training a model that fails to converge, the overhead of a few hours for permutation search is negligible. We view this search cost as a necessary investment to ensure training feasibility.

We will improve our work based on this valuable discussion.

Sincerely, Authors

---

### Note · Authors · 2025-12-03

I have read and agree with the venue's withdrawal policy on behalf of myself and my co-authors.